# A Study of Low Young’s Modulus Ti–15Ta–15Nb Alloy Using TEM Analysis

**DOI:** 10.3390/ma13245694

**Published:** 2020-12-14

**Authors:** Huey-Er Lee, Ju-Hui Wu, Chih-Yeh Chao, Yen-Hao Chang, Je-Kang Du, Ker-Kong Chen, Huey-Ming Chen

**Affiliations:** 1Department of Dentistry, Yuan’s General Hospital, Kaohsiung 80249, Taiwan; huerle@kmu.edu.tw; 2Department of Dentistry, Kaohsiung Medical University Hospital, Kaohsiung 80708, Taiwan; wujuhui1020@gmail.com; 3Department of Oral Hygiene, College of Dental Medicine, Kaohsiung Medical University, Kaohsiung 80708, Taiwan; 4Department of Mechanical Engineering, National Pintung University of Science and Technology, Pingtung 91201, Taiwan; cychai@mail.npust.edu.tw (C.-Y.C.); chmg@o-ta.com.tw (H.-M.C.); 5School of Dentistry, College of Dental Medicine, Kaohsiung Medical University, Kaohsiung 80708, Taiwan; edward590198@gmail.com

**Keywords:** Ti–15Ta–15Nb alloy, cast, microstructure, phase, kikuchi diagram analysis, orientation relationships

## Abstract

The microstructural characteristics and Young’s modulus of the as-cast Ti–15Ta–15Nb alloy are reported in this study. On the basis of the examined XRD and TEM results, the microstructure of the current alloy is essentially a mixture (α + β+ α′ + α″ + ω + H) phase. The new H phase has not previously been identified as a known phase in the Ti–Ta–Nb alloy system. On the basis of examination of the Kikuchi maps, the new H phase belongs to a tetragonal structural class with lattice parameters of a = b = 0.328 nm and c = 0.343 nm, denoting an optimal presentation of the atomic arrangement. The relationships of orientation between these phases would be {0001}_α_//{110}_β_//{$\bar{1}$2$\bar{1}$0}_ω_//{10$\bar{1}$}_H_ and (01$\bar{1}$0)_α_//(1$\bar{1}$2)_β_//($\bar{1}$010)_ω_//(121)_H._ Moreover, the Young’s modulus of the as-cast Ti–15Ta–15Nb alloy is approximately E = 80.2 ± 10.66 GPa. It is implied that the Young’s modulus can be decreased by the mixing of phases, especially with the presence of the H phase.

## 1. Introduction

Titanium and titanium alloys (Ti and Ti alloys) are used as biomaterials in medical implants because of their excellent biocompatibility, high corrosion resistance, and high strength-to-density ratio [1]. Titanium alloy (of which Ti–6Al–4V accounts for the majority) is used in approximately 70% to 80% of common total joint replacement surgery to replace human bone, such as the hip joint, and knee and shoulder prosthesis [2,3]. However, the major problem with the Ti alloy in medical implants is the stiffness mismatch with the surrounding bone. This leads to stress shielding that, in turn, can cause bone resorption, which leaves bones with an insufficient bearing load, causing implant failure [4]. For biomaterial applications, the Young’s modulus of the Ti-based implants that aid in replacing bone should be as low and as close to that of the human bone as possible (4–30 GPa) [1,5].

Researchers [6,7,8] have stated that the Young’s modulus of a Ti alloy decreases as the bond orders of Ti and the alloying atoms (Bo) and the metal d-orbital energy level (Md) increase along the metastable β phase boundary [9]. Thus, the Young’s modulus of Ti alloys can be decreased by controlling the presence of the metastable β phase with high Bo and Md [8,10,11]. Several metallic elements can serve as β phase stabilizers in Ti alloys, including Fe, V, Ta, Nb, Mo, Ni, Cr, and Cu. These stabilizers can also improve the mechanical properties and corrosion resistance of Ti-based implants [12,13,14,15]. Accordingly, a combination of different phases becomes one of the best ways to further reduce the Young’s modulus value. The phases include the hexagonal close-packed (hcp) crystal structure (α, α′ and ω phases), body-centered cubic, and orthorhombic crystal structures (β and α′′). Among these phases, the presence of the α or ω phase increases the Young’s modulus, owing to its higher Young’s modulus than that of the β phase. Therefore, it is expected that Ti consisting of β and α′′ phases would lead to a relatively low Young’s modulus in comparison with conventional commercial Ti alloys.

In addition to the low Young’s modulus, materials used in clinical settings must have an acceptable biocompatibility and perform their medical function without harming the human body. Ti–6Al–4V has been reported to cause toxicity and neurotoxicity in the body owing to the vanadium and aluminum content [16]. Recently, the addition of Mo, Ta, and Nb to Ti alloys has been extensively studied owing to their low Young’s modulus and nontoxicity [9,13,15,17,18,19]. An in vitro cytotoxicity test revealed the biocompatibility of Ti–Nb alloys to be better than culture plate and CP–Ti controls, indicating that the Ti–Nb alloy has excellent cytocompatibility and biological properties, such as being nontoxic and nonallergic [5,20]. On the other hand, the addition of Ta could result in a lower Young’s modulus as compared to the Ti–6Al–4V and CP–Ti alloys [21]. Furthermore, some ternary alloys, including Ti–Ta–Nb and Ti–Nb–Zr alloys, exhibit different structural compositions and an extremely low Young’s modulus. For example, Wei et al. [7] showed that the Ti–(15–30)Ta–10Nb alloys comprise different volume fractions of the α/α′, α″, and β phases after annealing, where the volume fraction of the α″ phase is observed to increase with increasing Ta. They proved that the lowest Young’s modulus is E = 60 GPa in the case of the Ti–30Ta–10Nb alloy. Liu [22] developed an α′ type Ti–15Nb–9Zr alloy with a low modulus of E = 39 GPa and a high strength of 850 MPa. Moreover, Meng [23] showed that a Ti–36Nb–5Zr alloy comprising (β + α″) phases can have a Young’s modulus of E = 48 GPa. In addition, the presence of the mixture (α″ + β + ω) phase in the Ti–5.3Mo–6.5Sn–10.2Nb–10Zr (wt.%) alloy can result in a Young’s modulus of E = 48 GPa [24]. Furthermore, Guo [25] developed a metastable β-type Ti–33Nb–4Sn (wt. %) alloy to achieve a reduced Young’s modulus of E = 36 GPa.

Because of its allotropic characteristics, a transformation from body-centered cubic (BCC) to hexagonal close-packed (HCP) martensite can be observed in the case of titanium and titanium alloys. It is well known that Ti-based alloys may produce a different Young’s modulus depending on their constituent phases. On the basis of previous studies, the formation of the martensite phase can be designed as an α′ phase belonging to the HCP structural class, with a = 0.295 nm and c = 0.468 nm [26], or an α″ phase as an orthorhombic structure, with a = 0.301–0.352 nm, b = 0.473–0.498 nm, and c = 0.453–0.496 nm [27,28,29]. Knowles and Smith and Qiu et al. [26,30] claimed that two morphologies of α′ martensite occurred: platelike martensite with parallel interfaces (Type I) and platelike martensite with a zig-zag twinning interface. In Type I, the α′ martensite is characterized by a habit plane of {344}β, which is perpendicular to the (0001)α′ basal plane of the HCP structure. The Type II α′ martensite also has a {344}β habit plane and contains {1$\bar{1}$01}α′ twins [26]. However, the orthorhombic α″ phase is known to occur while maintaining the lattice correspondences [100]α″//[100]β, [010]α″//[101]β, and [001]α″//[110]β [31]. The transformation strains from the β/α″ transformation seem to be primarily accommodated by an internal twinning on the {111}α″ planes. Furthermore, the transition of β to the hexagonal ω-phase confirms its three types of formation: (i) during quenching from the β phase in a high-temperature field [32]; (ii) through isothermal aging treatment at 100–500 °C [33,34]; (iii) a deformation-induced ω phase formed by applying stress [33,35]. The mechanism of the β–to–ω transformation has been proved as the ordered collapse along the {111}β lattice planes [36]. Meanwhile, the nonequilibrium phases (such as α′, α″, and ω) that may precipitate in the metastable β–Ti alloys via atomic arrangement under quenching can be controlled through material processing and/or an alloy-designed composition [37,38].

Although numerous combinations of Ti, Ta, and Nb alloys have developed by many researchers, there are still many compositions that need to be investigated and analyzed in order to obtain a lower Young’s modulus for Ti alloys. Furthermore, understanding the relationship between the composition, phases, and properties is an important area of study in this field. Therefore, this research aimed to investigate the microstructure and phase evolution of the promising Ti–15Ta–15Nb (mass%) alloy using SEM, XRD, and TEM observations. Micropillars of Ti–15Ta–15Nb alloy were formed to assess the Young’s modulus using a Nano Indenter.

## 2. Materials and Methods

### 2.1. Sample Preparation

The Ti–15Ta–15Nb alloy was obtained by melting high-purity Ti, Ta, and Nb elements (99.99%, 99.97%, and 99.99%, respectively, in mass%, Yingtai Metal Materials Co. Ltd., Dongguan, China) in a vacuum induction melting furnace at 1840 °C for 15 min, followed by cooling at a rate of approximately 3 °C/s in a copper crucible with a cooling cycle system. Thereafter, the ingot was re-melted three times to ensure compositional macrohomogeneity. The resulting composition (Ti–15.2Ta–14.8Nb–0.48Cu–0.08O–0.05N–0.03C in mass%) was determined by Inductively Coupled Plasma Atomic Emission Spectrometry (ICP–AES, PerkinElmer Inc.—Optima 2100 DV, Waltham, MA, USA). The alloy was purified in duplicate using a vacuum melting furnace to reduce segregation due to gravity during casting. After casting, the Ti–15Ta–15Nb ingots were cut into squared 10.0 × 10.0 × 2.0 (±0.05) mm sheets by wire cutting.

### 2.2. Young’s Modulus Analysis

We utilized a Nano Indenter G200 test system (Keysight, CA, USA) to analyze the Young’s modulus of the samples. It allows continuous contact stiffness measurements and the collection of load and displacement data at any point on the loading curve (not just at the unloading point), similar to that in conventional indentation measurements. The selected samples were machined by focus ion beam (FIB) milling to produce micropillars. The micropillars were 1 μm in diameter and 4 μm in height, with a taper angle of 1.6° and Ga damaged layers of 15 nm. The strain rate was set to approximately 10^−4^ s^−1^ to reach a maximum compressed load of 50 mN by flat tip [7]. Three Ti–15Ta–15Nb specimens were used to measure the Young’s modulus, with each specimen being randomly measured at 48 points. Finally, the Young’s modulus of the samples was calculated from 144 values.

### 2.3. Microstructural Analysis

We analyzed the microstructures via scanning electron microscope (SEM, JSM–6380, JEOL, Japan), transmission electron microscope (TEM, Philips CM200 FEG, Eindhoven, The Netherlands), and X-ray diffractometry (XRD, Bruker D8, Billerica, MA, USA), as follows. For the SEM analysis, the samples were fixed using an epoxy hardener and a cold mounting press, followed by sequential grinding with 100, 600, 1000, and 1500 grit waterproof abrasive papers (MATADOR, Remscheid, Germany). Then, the samples were polished with 0.3-μm alumina powder, followed by secondary polishing with silicon dioxide, and water polishing to remove surface impurities. Finally, they were treated with a corrosive fluid containing 7.5 vol.% HNO_3_ (70%, Sigma–Aldrich, St. Louis, MO, USA), 2.5 vol.% HF (48%, Sigma–Aldrich, St. Louis, MO, USA), and 90 vol. % alcohol (96%, Sigma–Aldrich, St. Louis, MO, USA). The surface structural properties were observed using SEM with an operating voltage of 15 kV and a vacuum of 10^−4^ Pa. In the case of the SEM, backscattered electrons were detected.

For the TEM observations, the samples were polished with water sandpaper (80 μm) and then electropolished with an automatic twin-jet electropolisher (Model 110, Fischion Instrument, PA, USA) at a working voltage of 30–40 V and a temperature range from approximately −15 to −5 °C using 85 vol.% methanol (CH_3_OH, 99.8%, Sigma–Aldrich, St. Louis, MO, USA) + 15 vol.% perchloric acid (HClO_4_, 70%, Sigma–Aldrich, St. Louis, MO, USA) as the corrosive fluid. Bright-field (BF) and dark-field (DF) images were obtained using a TEM system, and selected area diffraction patterns (SADPs) of the samples were also recorded. Kikuchi lines were shaped in the diffraction patterns by diffusely scattered electrons because of the thermal atom vibrations [39]. A detailed analysis of the new phase was used to build the Kikuchi map using various SADPs. To determine the composition of each phase, energy-dispersive spectroscopy (EDS) was conducted using a field-emission TEM system (Tecnai F20 G2 FEI, Eindhoven, the Netherlands).

XRD characterization was performed using a diffractometer (Bruker D8, Billerica, MA, USA) with Cu Kα radiation (λ = 1.5406 Å) at 40 kV, 30 mA, and room temperature. The scanning range was 20°–80° at a step of 0.1° and a holding time of 5 s.

## 3. Results

Figure 1 shows three SEM electron micrographs of the as-cast Ti–15Ta–15Nb alloy. The microstructures exhibit equiaxed grains with needle martensitic precipitates with a length of 2–6 μm and a width of 1–2 μm (Figure 1b,c). Figure 2 shows a BF TEM electron micrograph at low magnification. It shows the occurrence of various platelike precipitates, islandlike precipitates, and a matrix with dislocation morphology. Figure 3a is a BF electron micrograph captured from area A marked in Figure 2. Figure 3b,c shows two SADPs captured from the areas marked α and α′, respectively; the foil normal is g = [11$\bar{2}$0]. It also shows that the α and α′ precipitates belong to an HCP structural class, with lattice parameters a = 0.294 nm and c = 0.471 nm and no significant difference between α and α′, which is consistent with the XRD results. Moreover, the figure shows different intensity characteristics in the g = 0001 reflection spot. The intensity of the reflection spots is influenced by structural factors and shows that the atomic position and atomic scattering factor influence the intensity of the spots. The equation for the structural factor is as follows
(1)$$F_{hkl}=\overset{N}{\underset{i=1}{\sum }}f_{n}e^{2\pi i\left(hu_{n}+kv_{n}+lw_{n}\right)}$$
where *f_n_* is the atomic scattering factor, (*h,k,l*) is the Miller index, and (*u,v,w*) is the atomic position [40].

Figure 4a is a BF electron micrograph captured from area B marked in Figure 2. Figure 4b,c shows two SADPs captured from area B. In the figures, the reflection spots of the α, β, and ω phases are shown. The lattice parameter of the β phase is a = 0.327 nm and those of the ω phase are a = 0.468 nm and c = 0.294 nm. These observations are similar to those of a previous study [41]. Additionally, the extra reflection spots (yellow arrow in Figure 4c) could not be ascribed to any structure of the α, β, or ω phases. To clarify whether the extra spot is double diffraction, DF electron micrographs were obtained. Figure 4d–f show three DF electron micrographs. These show that the phases can be individually observed. This also means that the extra spots are not the result of double diffraction. In this study, we denote these extra diffraction spots as the H phase. As shown in Figure 5a–c, the chemical compositions of the β, ω, and H phases are Ti–21.2 wt. % Ta–22.3 wt. % Nb, Ti–22.8 wt. % Ta–23.1 wt. % Nb, and Ti–13.5 wt. % Ta–11.8 wt. % Nb, respectively. Furthermore, the β and H phases exhibit a similar composition, and the ω phase has a lower Ta and Nb. It was of interest to determine whether the structure of this new phase was that of the α” phase. Therefore, an analysis of the Kikuchi line map was performed, as shown in Figure 6a,b; the foil normal and related angles are shown, respectively. We arranged the SADPs of the different axes ([1], [12], [11], [$\bar{1}$33], [$\bar{1}$11], and [$\bar{1}$13]) and ([12], [0$\bar{1}$1], [$\bar{1}$33], [$\bar{1}$11], and [$\bar{1}$23]) by following the Kikuchi line rule to verify the properties of this new phase. Furthermore, the d-spacing of the reflection spots was measured, as presented in Table 1. Moreover, the nanobeam diffraction of the [$\bar{1}$11] pole was examined, as shown in Figure 6c. It shows that a two-fold symmetrical hexagon image occurred within the reflection spots. This result cannot be used to determine if this new phase belongs to the α” phase or another phase that was previously discovered. On the basis of the H phase Kikuchi line map, it was found that the lattice constants of this phase are a = b = 0.328 nm and c = 0.343 nm, indicating that the lattice constant in one of the directions is larger than the typical lattice constant of BCC. The atomic arrangement of the H phase is similar to that of a stretched BCC structure. In the current analysis, it is optimal to determine whether the H phase belongs to a body-centered tetragonal (BCT) structural class. An indexed schematic diagram of Figure 4c is shown in Figure 7. Meanwhile, the orientation relationship between the α, β, ω, and H phases is {0001}_α_//{110}_β_//{$\bar{1}$2$\bar{1}$0}_ω_//{10$\bar{1}$}_H_ and (01$\bar{1}$0)_α_//(1$\bar{1}$2)_β_//($\bar{1}$010)_ω_//(121)_H_.

Figure 8 presents the representative indexed XRD patterns. It shows that the α, α″, and ω phases and the metastable β phase can be detected. Additionally, the diffraction peaks in the H phase were indexed as blue lines, as shown in Figure 8a. Two local XRD patterns show that three peaks could be ascribed to the H phase, as shown in Figure 8b,c.

The mean Young’s modulus of as-cast Ti–15Ta–15Nb alloy is 80.2 ± 10.66 GPa. The typical values and the distribution of the Young’s modulus are shown in Figure 9. It was found that the Young’s modulus of the as-cast Ti–15Ta–15Nb alloy is within the range 57.3–108.5 GPa and has a Bell-shaped distribution. Most of the Young’s modulus values are between 70 and 90 GPa.

## 4. Discussion

It is well known that TEM analysis has a very small range for the local observation of microstructures and phases. To clarify the microstructure of the present alloy, an XRD analysis was performed, as shown in Figure 8. The reflection peaks are similar to those of previous studies on Ti–Ta, Ti–Nb, and Ti–Ta–Nb systems. This shows that the nonequilibrium phases, i.e., the α, α″, and ω phases and metastable β phase, can be detected [7,42,43]. The 2θ angles of the diffraction peaks in different H phase planes are indicated with blue lines in Figure. 8a. There are three peaks that can be ascribed to the H phase, as shown in Figure 8a,b. This indicated that the H phase could be found in a large observation area.

Figure 4b,c show various extra peaks that had not been identified before and belong to (110)_H_, (012)_H_, and (112)_H_. Therefore, the microstructure of the present as-cast alloy is essentially a mixture (α + α′+ α″ + β + ω + H) phase. On the basis of the transition of β→α, under different cooling rate and composition conditions, the α′ and α″ phases are formed via rapid quenching or the lower activity-energy barrier from the β field and are referred to as martensite. The volume change of β→α, which decreases to Δ = 0.738%, implies that the energy barrier of martensitic transformation on the present alloy is low. In the present study, the c/a ratio of the α or α′ phase is 1.602, which is slightly higher than the c/a ratio of 1.586 found in other observations [7,44]. Moreover, the formation of the ω phase can be observed by either quenching from the β field or during the aging of the quenched Ti alloys. According to Kikuchi maps, the H phase can be ascribed to a BCT structural class with lattice parameters a = b = 0.328 nm and c = 0.343 nm. We believe it to be a BCC structure with distortion owing to a stretching force along the <010> direction. Furthermore, the composition of the H phase is also similar to that of the β phase. Notably, the H phase has not been previously reported.

The typical values and the distribution of Young’s modulus are shown in Figure 9. The Young’s modulus of the as-cast Ti–15Ta–15Nb alloy is within the range 57.3–108.5 GPa, and its average is 80.2 ± 10.66 GPa, which makes it favorable for biomaterial applications as compared to commercial pure Ti and/or Ti–6Al–4V–ELI alloys (listed in Table 2). A lower Young’s modulus decreases the stress-shielding effect in implants. This shows that Ti alloys with multiple phases (β and α″ phases) can decrease the Young’s modulus. However, the formation of the ω phase increases the strength and Young’s modulus [11]. Moreover, Wei et al. [7] reported that the Young’s modulus and microstructure of the Ti–15Ta–10Nb alloy is approximately E = 90 GPa with a (α + β + ω + α″) phase. In contrast, the as-cast Ti–15Ta–15Nb alloy fabricated in the present study consists of the α, β, ω, α″, and H phases, which demonstrates that the presence of the H phase may further decrease this parameter.

## 5. Conclusions

The Young’s modulus of the Ti–15Ta–15Nb alloy in the as-cast condition is E = 80.2 ± 10.66 GPa, which identifies it as having a great potential as a biomaterial and in implants. Furthermore, the main purpose of the present study was to investigate the microstructure of the as-cast Ti–15Ta–15Nb alloy. On the basis a series of analyses, the following conclusions were drawn:On the basis the XRD and TEM examinations, the microstructure of the as-cast Ti–15Ta–15Nb alloy is a mixture (α + β + α′ + α″ + ω + H) phase;The α and α′ phases belong to the HCP structural class with the lattice parameters a = 0.294 nm and c = 0.471 nm and c/a = 1.602, which are slightly higher than 1.586. The observation of the BCC β phase, orthorhombic α″, and hexagonal ω phase is similar to the results obtained by other researchers;A new phase, denoted the H phase, was observed. The H phase belongs to the tetragonal structural class with the lattice parameters a = b = 0.328 nm and c = 0.343 nm;The orientation relationship between these phases of the as-cast Ti–15Ta–15Nb alloy is {0001}_α_//{110}_β_//{$\bar{1}$2$\bar{1}$0}_ω_//{10$\bar{1}$}_H_ and (01$\bar{1}$0)_α_//(1$\bar{1}$2)_β_//($\bar{1}$010)_ω_//(121)_H._

## Figures and Tables

**Figure 1 materials-13-05694-f001:**
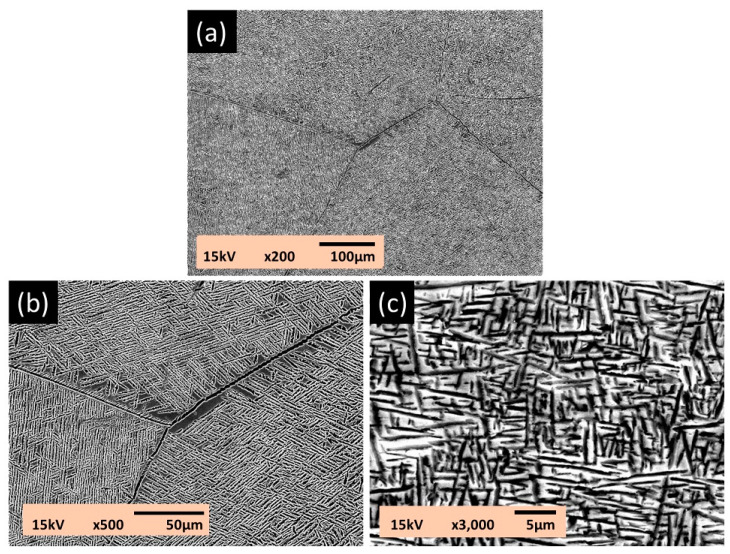
SEM electron micrographs of the as-cast Ti–15Ta–15Nb alloy: (**a**) ×200, (**b**) ×500, and (**c**) ×3000 magnification.

**Figure 2 materials-13-05694-f002:**
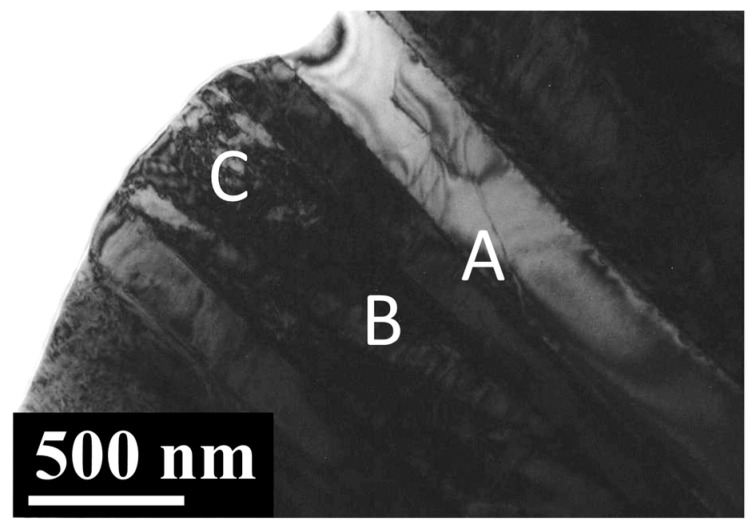
The bright-field (BF) TEM electron micrograph of the as-cast Ti–15Ta–15Nb alloy, which indicates the observed area. A, B and C indicating 3 different regions for following TEM analysis.

**Figure 3 materials-13-05694-f003:**
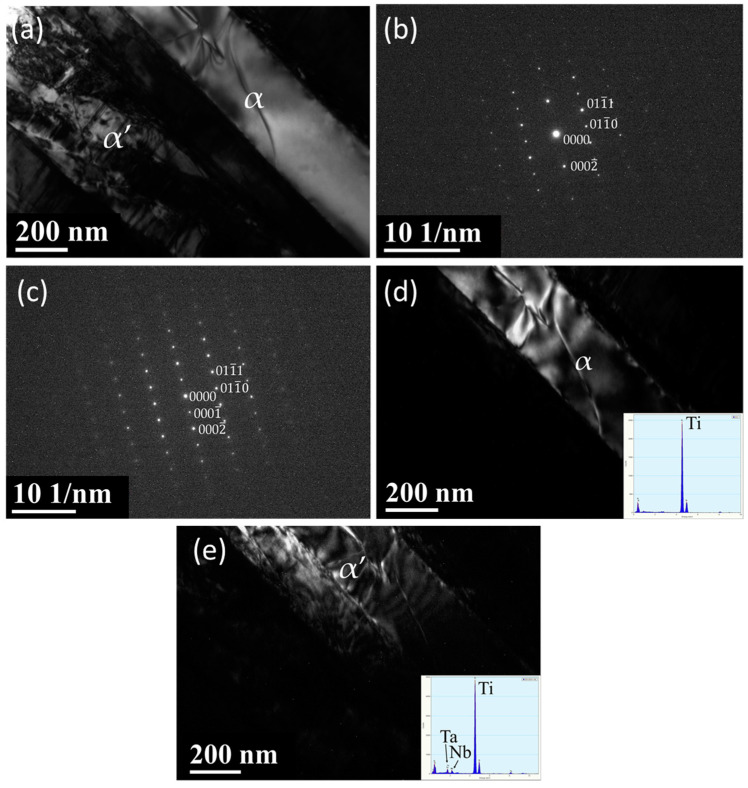
TEM electron micrographs of the as-cast Ti–15Ta–15Nb alloy captured from the area marked A in Figure 2. (**a**) BF; (**b**) α–phase selected area diffraction patterns (SADP), with a foil normal of [11,12,13,14,15,16,17,18,19,20], showing a weaker g = 0001 reflection; (**c**) α′–phase SADP, with a foil normal of [11,12,13,14,15,16,17,18,19,20], showing a stronger g = 0001 reflection; and (**d**,**e**) demonstrate two dark fields (DFs) showing the occurrence of α and α′ phases, respectively. TEM-energy-dispersive spectroscopy (EDS) data are also shown in the figure.

**Figure 4 materials-13-05694-f004:**
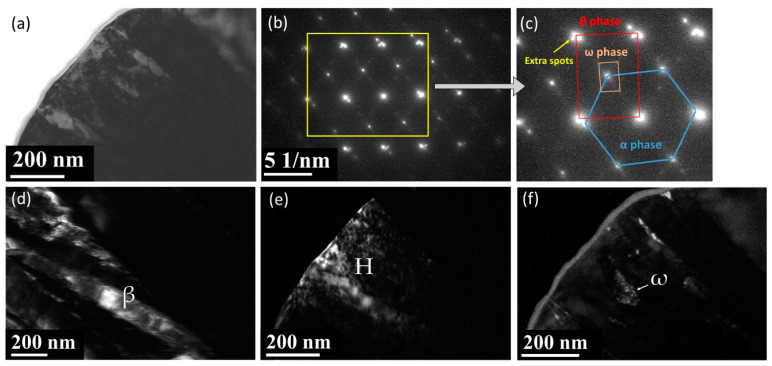
TEM electron micrographs of the as-cast Ti–15Ta–15Nb alloy captured from the area marked as B in Figure 2. (**a**) BF; (**b**,**c**) show two SADPs with the reflection spots of α, H, β, and ω phases; (**d**–**f**) three DFs showing the occurrence of the β, H, and ω phases, respectively.

**Figure 5 materials-13-05694-f005:**
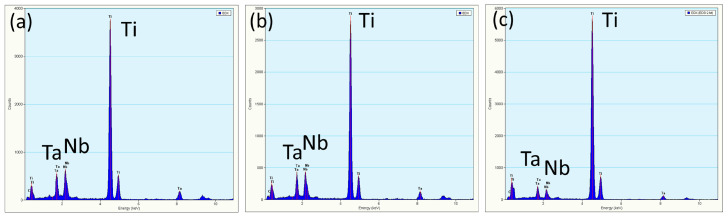
EDS analysis results for the phases in Figure 4d–f. (**a**) The β phase; (**b**) H phase; and (**c**) ω phase.

**Figure 6 materials-13-05694-f006:**
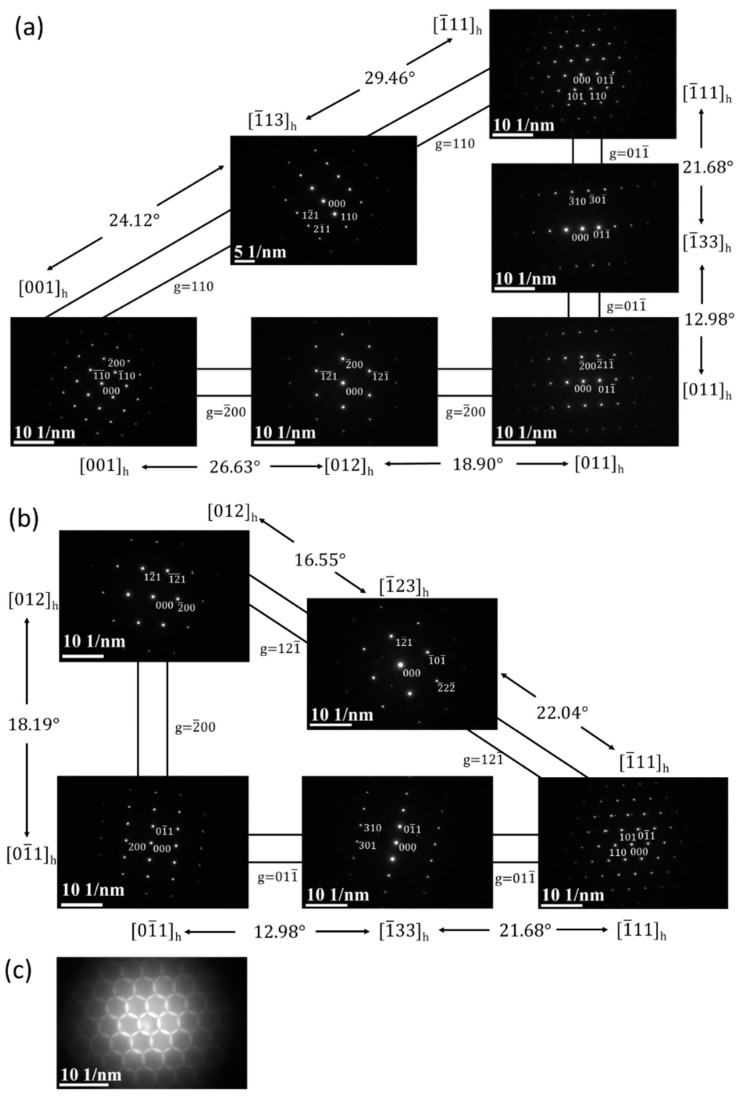
SADP electron micrographs captured from the H phase matrix: (**a**,**b**) show two Kikuchi maps for the crystal structure of the H phase; and (**c**) shows a nanobeam diffraction pattern, with a foil normal of [$\bar{1}$11] pole.

**Figure 7 materials-13-05694-f007:**
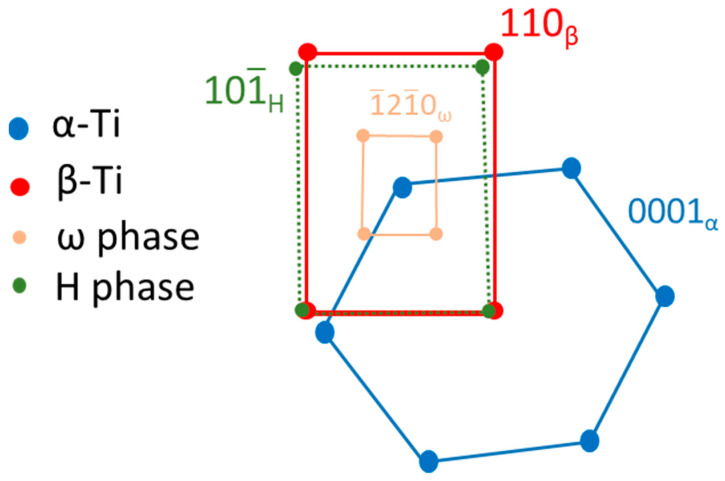
Schematic diagram illustrating the indexed diffractogram.

**Figure 8 materials-13-05694-f008:**
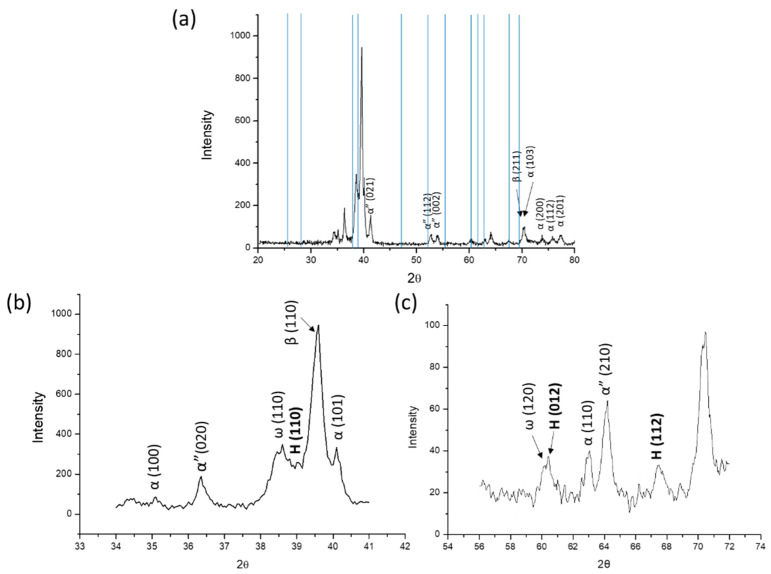
(**a**) XRD patterns of the as-cast Ti–15Ta–15Nb alloy (blue lines show the diffraction peaks of H phase). (**b**,**c**) show the magnification of the local diffraction angle from (**a**).

**Figure 9 materials-13-05694-f009:**
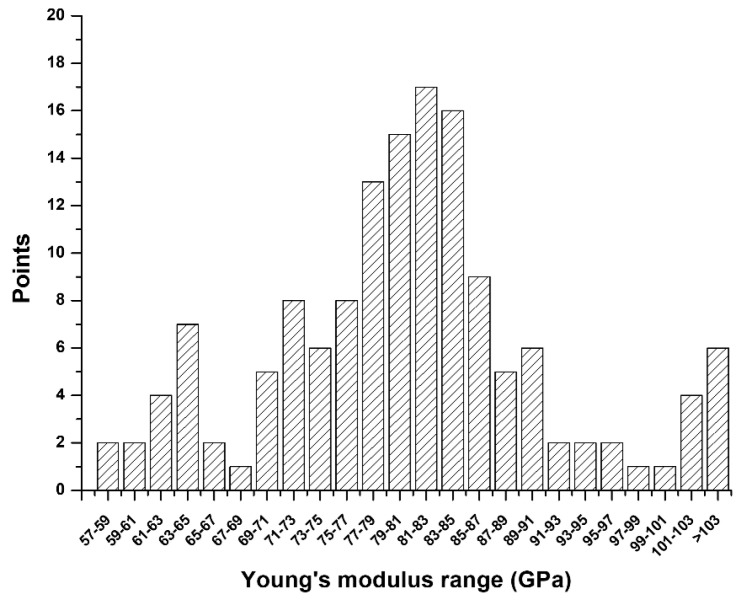
Typical values and distribution of the Young’ modulus of the as-cast Ti–15Ta–15Nb alloy.

**Table 1 materials-13-05694-t001:** D-spacings of the H phase.

Number	Observed D-Spacing *	Calculated D-Spacing **	Indexed Plane
1	0.3284	0.3284	100
2	0.3284	0.3284	010
3	0.2372	0.2372	011
4	0.2362	0.2372	0$\bar{1}$1
5	0.2372	0.2372	101
6	0.2321	0.2322	110
7	0.1922	0.1922	$\bar{1}$11
8	0.1924	0.1922	1$\bar{1}$1
9	0.1352	0.1350	211
10	0.1350	0.1350	2$\bar{1}$. 1
11	0.1348	0.1350	$\bar{2}$1$\bar{1}$
	0.1352	0.1350	1$\bar{2}$1
13	0.1350	0.1350	$\bar{1}$21
14	0.1040	0.1038	$\bar{3}$10
15	0.1038	0.1038	$\bar{13}$0
16	0.1042	0.1042	$\bar{3}$0$\bar{1}$
17	0.1044	0.1042	$0\bar{3}$1
18	0.0980	0.0960	$2\bar{2}$2

* Derived from the selected area diffraction patterns. ** Based on the body a = 0.328 nm and c = 0.343 nm.

**Table 2 materials-13-05694-t002:** Young’s modulus of the Ti and Ti−6Al−4V alloys and the Ti–Ta–Nb alloy system.

Alloys	E (GPa)	Phases	Treatment	Ref.
Pure Ti grade 2	102.7	α	-	[45]
Ti−6Al−4V ELI	110–114	α + β	450 °C–650 °C	[45]
Ti−15Ta−15Nb	80.2	α + α′ + α″ + β + ω + H	As-cast	This work

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
