# Peer review of "A Study of Low Young’s Modulus Ti–15Ta–15Nb Alloy Using TEM Analysis"

_materials, 2020, doi:10.3390/ma13245694_

Round 1

Reviewer 1 Report

  1. This is an interesting paper that concerns the microstructure and Young’s modulus of a novel Ti-15Ta-15-Nb alloy. The title erroneously implies that more than one alloy is being investigated. It is a shame that the authors haven’t examined the microstructure as a function of different cooling rates, as this would have been of additional interest to the readers of this journal.

  1. In the introduction, it would be nice if the authors would explain the rational for studying the Ti alloy of this particular composition. I can guess that it’s because Ti-Nb alloys exhibit favourable biocompatibility with various human tissues and that Ti-Ta alloys have appropriately low Young’s moduli; but I would like the authors to confirm this or otherwise justify their curiosity in this specific alloy.

  1. The following sentences, at the end of the introduction, are misleading and need to be removed, as this study neither considers the biocompatibility of the Ti-15Ta-15-Nb alloy nor investigates its microstructural and mechanical properties as a function of composition:

‘Therefore, this research aims to develop a new class of Ti–15Ta–15Nb alloys that exhibit a low Young's modulus and focus on the microstructures by TEM observations. Meanwhile, as a new biomaterial, the biocompatibility of the present alloy is also be evaluated.’

These may be the broader intentions of the authors, but they are not the aims and objectives of the study that is reported in this manuscript.

  1. It would be good for the authors to provide a more comprehensive summary of the main objectives of the study at the end of the introduction that briefly summarises the methods that have been used. This would enable readers who are already familiar with this field of research to skip to the results section and only refer back to the materials and methods if they need to check certain parameters.

  1. The reagent supplier and the melting times and temperatures (including cooling rates) should be given in the first paragraph of the materials and methods section. The method for determining the composition of the alloy should also be given in this section. Into what medium were the ingots cast?

  1. Figures 7 and 8 contain results that should be reported in the results section, not the discussion. The mean Young’s modulus should also be reported in the results section.

  1. It is easy to understand the content of the manuscript and the information is presented in a logical order; but, the grammar needs to be corrected by a native English speaker and the text needs to be formatted consistently.

I have no confidential comments for the editor.

Author Response

06/12/2020

Dr. Josef Stráský

Charles University, Department of Physics of Materials, Prague, Czech Republic

Guest Editor

Materials

Dear Prof. Dr. Josef Stráský:

    We would like to resubmit our revised manuscript, titled " A study of low Young’s modulus Ti–15Ta–15Nb alloys using TEM analysis " (Manuscript ID: materials-1010951), for consideration for publication in Materials.

   The editor and reviewers have recommended minor revisions. We have carefully reviewed these suggestions and revised our text accordingly. Please note that those words within manuscript, tables and figures are written with red words for correction. Our responses to each of the reviewers’ comments are detailed below. The details are provided in the manuscript and mark with red for correction.

Comments and Suggestions for Authors:

Review 1

1. This is an interesting paper that concerns the microstructure and Young’s modulus of a novel Ti-15Ta-15-Nb alloy. The title erroneously implies that more than one alloy is being investigated. It is a shame that the authors haven’t examined the microstructure as a function of different cooling rates, as this would have been of additional interest to the readers of this journal.

Answer: Thank you for the reviewer’s suggestions. We have modified the word from “alloys” to “alloy”. Cooling rates is one of interest topic especially for a new Ti-Ta-Nb alloy. However, we focused on the characteristic between microstructural analysis and Young’s modulus of the as-cast Ti-15Ta-15Nb alloy in this study. Among the study, we have found that a special phase which can not be identified as known  phase in Ti alloy system. Thus, a series of TEM analysis have been conducted. We believed that the present study can provide readers with certain information and help in the analysis of titanium alloy microstructure especially Kikuchi line map analysis. 

2. In the introduction, it would be nice if the authors would explain the rational for studying the Ti alloy of this particular composition. I can guess that it’s because Ti-Nb alloys exhibit favourable biocompatibility with various human tissues and that Ti-Ta alloys have appropriately low Young’s moduli; but I would like the authors to confirm this or otherwise justify their curiosity in this specific alloy.

Answer: Thank you for your kind suggestion. We have added the description in the introduction based on the reviewer’s suggestion at line 62-66 Page 2 of revised manuscript and marked with red.

3. The following sentences, at the end of the introduction, are misleading and need to be removed, as this study neither considers the biocompatibility of the Ti-15Ta-15-Nb alloy nor investigates its microstructural and mechanical properties as a function of composition:

Therefore, this research aims to develop a new class of Ti–15Ta–15Nb alloys that exhibit a low Young's modulus and focus on the microstructures by TEM observations. Meanwhile, as a new biomaterial, the biocompatibility of the present alloy is also be evaluated.’

These may be the broader intentions of the authors, but they are not the aims and objectives of the study that is reported in this manuscript.

Answer: We appreciate this helpful comment from the reviewer. The purposes of the present study have been rewritten to be aims and objectives of the study. This information has been supplemented at line 102-105 Page 3 of revised manuscript and marked with red.

4. It would be good for the authors to provide a more comprehensive summary of the main objectives of the study at the end of the introduction that briefly summarises the methods that have been used. This would enable readers who are already familiar with this field of research to skip to the results section and only refer back to the materials and methods if they need to check certain parameters.

Answer: Thank you for your pointing out. At the end of introduction, a brief summary of the methods that have been used in this study was presented. This information has been supplemented at line 102-105 Page 3 of revised manuscript and marked with red.

5. The reagent supplier and the melting times and temperatures (including cooling rates) should be given in the first paragraph of the materials and methods section. The method for determining the composition of the alloy should also be given in this section. Into what medium were the ingots cast?

Answer: We appreciate this helpful comment from the reviewer. The reagent supplier, the melting times, temperatures including cooling rates and cast medium were added in the first paragraph of the materials and methods section. This information has been supplemented at line 108-112 Page 3 of revised manuscript and marked with red.

The method for determining the composition of the alloy was performed by ICP-AES. This information has been supplemented at line 112-114 Page 3 of revised manuscript and marked with red.

6. Figures 7 and 8 contain results that should be reported in the results section, not the discussion. The mean Young’s modulus should also be reported in the results section.

Answer: Thank you for your pointing out. We have changed Figure 7 into Figure 8 and previous Figure 8 modified into Figure 9 in the results section. The mean Young’s modulus should also be reported in the results section. This information has been supplemented at line 228-231 and 235-238 Page 3 of revised manuscript and marked with red.

7. It is easy to understand the content of the manuscript and the information is presented in a logical order; but, the grammar needs to be corrected by a native English speaker and the text needs to be formatted consistently.

Answer: Thank you for your comments and suggestions. The manuscript has been edited by a professional English editing group to check grammar error. Besides, we have attached the CERTIFICATE OF ENGLISH EDITING.

Thank you for your consideration. We hope our manuscript is suitable for publication in your journal.

Sincerely,

Je-Kang Du

Associate professor, School of Dentistry, Kaohsiung Medical University

100 Shih-Chuan 1st Road, San-Ming District, Kaohsiung, Taiwan 807

Tel.: + 886-7-3121101 ext. 7006, Fax: + 886-7-3121510

E-mail: dujekang@gmail.com

Reviewer 2 Report

The article is interest to reader, but need changes described below.

The proposed changes will increase its quality

Remarks:

Line 108: Authors presented the results of composition, but without information about the method used to measure the chemical composition,

Line: 122: There is no imformation about SEM. Informations abour SEM from line 129 should be replaced to line 122.

Lines 134-135: There is no information about used chemical reagents such as manufacter, purity

Lines 147-149: The description should be deleted.

In “Results” section Young's modulus measurement results are missing. This information should be moved from the "Discussion" section

There is a lack of information on the number of measurements of mechanical properties. What was the repeatability of the obtained results? The results should be presented with standard deviations.

Author Response

06/12/2020

Dr. Josef Stráský

Charles University, Department of Physics of Materials, Prague, Czech Republic

Guest Editor

Materials

Dear Prof. Dr. Josef Stráský:

    We would like to resubmit our revised manuscript, titled " A study of low Young’s modulus Ti–15Ta–15Nb alloys using TEM analysis " (Manuscript ID: materials-1010951), for consideration for publication in Materials.

   The editor and reviewers have recommended minor revisions. We have carefully reviewed these suggestions and revised our text accordingly. Please note that those words within manuscript, tables and figures are written with red words for correction. Our responses to each of the reviewers’ comments are detailed below. The details are provided in the manuscript and mark with red for correction.

Comments and Suggestions for Authors:

Reviewer 2

Comments and Suggestions for Authors

The article is interest to reader, but need changes described below.

The proposed changes will increase its quality

Remarks:

Line 108: Authors presented the results of composition, but without information about the method used to measure the chemical composition,

Answer: Thank you for your pointing out. The method for determining the composition of the alloy was performed by ICP-AES. This information has been supplemented at line 112-114 Page 3 of revised manuscript and marked with red. 

Line: 122: There is no information about SEM. Information about SEM from line 129 should be replaced to line 122.

Answer: We appreciate this helpful suggestion from the reviewer. The information of SEM was replaced as shown at line 129 Page 3 of revised manuscript and marked with red.    

Lines 134-135: There is no information about used chemical reagents such as manufactur, purity

Answer: Thank you for your deep concern. This information has been supplemented at line 135-137 and 143 Page 3 of revised manuscript and marked with red.

Lines 147-149: The description should be deleted.

Answer: Thank you for your suggestion. The description has been deleted.

In “Results” section Young's modulus measurement results are missing. This information should be moved from the "Discussion" section

Answer: Thank you for your deep concern. We have reported XRD and Young's modulus results in the results section at line 235-238 Page 9 of revised manuscript and marked with red.

There is a lack of information on the number of measurements of mechanical properties. What was the repeatability of the obtained results? The results should be presented with standard deviations.

Answer: Thank you for your pointing out. We have reported the number of measurements of mechanical properties. We randomly tested Ti-15Ta-15Nb alloy in large quantities to reduce the deviation, and use the average value as the Young’s modulus of the alloy. These data show a bell-shaped distribution (Normal distribution) as shown in Fig. 9 of revised manuscript indicating that they meet a strict and stable probability distribution. The Young’s modulus has been presented with standard deviations.

Thank you for your consideration. We hope our manuscript is suitable for publication in your journal.

Sincerely,

Je-Kang Du

Associate professor, School of Dentistry, Kaohsiung Medical University

100 Shih-Chuan 1st Road, San-Ming District, Kaohsiung, Taiwan 807

Tel.: + 886-7-3121101 ext. 7006, Fax: + 886-7-3121510

E-mail: dujekang@gmail.com

Reviewer 3 Report

This is a purely experimental paper, dealing with an (in-principle) interesting topic. It presents interesting results and findings, which can be useful to the research community. As such, it has publication merit, but a revision needs to be made to address the following minor issues:

Abstract:

  • Very hard to read, primarily due to the notation and symbology used. Please rewrite.
  1. Introduction

(NOTE: overall well written, providing good background information)

  • “…wieldy used…”: Not an appropriate expression (‘wieldy’). Also, please check for other (minor) grammar issues (I have found a few)
  1. Materials and Methods

(NOTE: overall pleased with the quality of the information provided)

  • Can you please justify the selection of this strain rate value (section 2.2)? Has this been used by other researhers?
  1. Results

(NOTE: overall well written and very detailed)

  • Please delete this (it is copied from the template I presume): “This section may be divided by subheadings. It should provide a concise and precise description of the experimental results, their interpretation as well as the experimental conclusions that can be drawn.”
  • The clarity of Fig. 1 has to be improved, also the authors may consider adding another micrograph with a magnification between 100μm and 5μm (to show the features)
  • Can you please elaborate more on this? “Based on the Kikuchi line map of the H phase, the atomic arrangement is similar to that of a stretched BCC structure.”
  • The graphs embedded in Fig. 4(d), (e) and (f) are too small – please increase their size or put them on separate figure
  1. Discussion

(NOTE: The section discusses successfully the obtained results)

  • Figure 8: A narrower frequency diagram may be needed – please revise or add
  1. Conclusions

Very good

Author Response

06/12/2020

Dr. Josef Stráský

Charles University, Department of Physics of Materials, Prague, Czech Republic

Guest Editor

Materials

Dear Prof. Dr. Josef Stráský:

    We would like to resubmit our revised manuscript, titled " A study of low Young’s modulus Ti–15Ta–15Nb alloys using TEM analysis " (Manuscript ID: materials-1010951), for consideration for publication in Materials.

   The editor and reviewers have recommended minor revisions. We have carefully reviewed these suggestions and revised our text accordingly. Please note that those words within manuscript, tables and figures are written with red words for correction. Our responses to each of the reviewers’ comments are detailed below. The details are provided in the manuscript and mark with red for correction.

Comments and Suggestions for Authors:

Reviewer 3

This is a purely experimental paper, dealing with an (in-principle) interesting topic. It presents interesting results and findings, which can be useful to the research community. As such, it has publication merit, but a revision needs to be made to address the following minor issues:

Abstract:

Very hard to read, primarily due to the notation and symbology used. Please rewrite.

Answer: Thank you for your deep concern. The abstract has been rewritten. The lengthy details and symbology have been modified.

Introduction

(NOTE: overall well written, providing good background information)

Answer: Thank you for your comments.

“…wieldy used…”: Not an appropriate expression (‘wieldy’). Also, please check for other (minor) grammar issues (I have found a few)

Answer: Thank you for your pointing out. “…wieldy used…” has been modified. Additionally, the manuscript has been edited by a professional English editing group to check grammar error. Besides, we have attached the CERTIFICATE OF ENGLISH EDITING.

Materials and Methods

(NOTE: overall pleased with the quality of the information provided)

Answer: Thank you for your comments.

Can you please justify the selection of this strain rate value (section 2.2)? Has this been used by other researhers?

Answer: Thank you for reviewer’s comments. The strain rate value has been cited an specific reference (Reference 7 of revised manuscript) which we followed. The reference is: Wei, T.Y.; Huang, J.C.; Chao, C.-Y.; Wei, L.L.; Tsai, M.T.; Chen, Y.H. Microstructure and elastic modulus evolution of TiTaNb alloys. J. Mech. Behav. Biomed. Mater. 2018, 86, 224-231.

Results

(NOTE: overall well written and very detailed)

Answer: Thank you for your comments.

Please delete this (it is copied from the template I presume): “This section may be divided by subheadings. It should provide a concise and precise description of the experimental results, their interpretation as well as the experimental conclusions that can be drawn.”

Answer: Thank you for your kind suggestion. The description has been deleted.

The clarity of Fig. 1 has to be improved, also the authors may consider adding another micrograph with a magnification between 100μm and 5μm (to show the features)

Answer: Thank you for your suggestion to make our data clear. A ×500 magnification SEM image was supplemented in figure 1 of revised manuscript.

Can you please elaborate more on this? “Based on the Kikuchi line map of the H phase, the atomic arrangement is similar to that of a stretched BCC structure.”

Answer: We appreciate this helpful comment from the reviewer. It has been elaborated more as shown in line 202-207 of revised manuscript and marked with red.

The graphs embedded in Fig. 4(d), (e) and (f) are too small – please increase their size or put them on separate figure

Answer: Thank you for your suggestion to make our data clear. A separate figure (Figure 5) was used to present the EDS results in revised manuscript.

Discussion

(NOTE: The section discusses successfully the obtained results)

Figure 8: A narrower frequency diagram may be needed – please revise or add

Answer: We appreciate this helpful comment from the reviewer. A new narrower frequency diagram of Young’s modulus range was made as shown in Figure 9 in revised manuscript. 

  1. Conclusions

Very good

Answer: Thank you for your comments.

Thank you for your consideration. We hope our manuscript is suitable for publication in your journal.

Sincerely,

Je-Kang Du

Associate professor, School of Dentistry, Kaohsiung Medical University

100 Shih-Chuan 1st Road, San-Ming District, Kaohsiung, Taiwan 807

Tel.: + 886-7-3121101 ext. 7006, Fax: + 886-7-3121510

E-mail: dujekang@gmail.com
